# Quality Attributes of Cryoconcentrated Calafate (*Berberis microphylla*) Juice during Refrigerated Storage

**DOI:** 10.3390/foods9091314

**Published:** 2020-09-18

**Authors:** Patricio Orellana-Palma, Guisella Tobar-Bolaños, Nidia Casas-Forero, Rommy N. Zúñiga, Guillermo Petzold

**Affiliations:** 1Department of Biotechnology, Universidad Tecnológica Metropolitana, Las Palmeras 3360, P.O. Box, 7800003 Ñuñoa, Santiago, Chile; rommy.zuniga@utem.cl; 2Laboratory of Cryoconcentration, Department of Food Engineering, Universidad del Bío-Bío, Av. Andrés Bello 720, Casilla 447, 3780000 Chillán, Chile; guisella.tobar1901@alumnos.ubiobio.cl (G.T.-B.); nidia.casas1701@alumnos.ubiobio.cl (N.C.-F.); gpetzold@ubiobio.cl (G.P.); 3Magíster en Ciencias e Ingeniería en Alimentos, Universidad del Bío-Bío, Av. Andrés Bello 720, Casilla 447, 3780000 Chillán, Chile; 4Doctorado en Ingeniería de Alimentos, Universidad del Bío-Bío, Av. Andrés Bello 720, Casilla 447, 3780000 Chillán, Chile; 5Programa Institucional de Fomento a la I+D+i, Universidad Tecnológica Metropolitana, Ignacio Valdivieso 2409, San Joaquín, 8940577 Santiago, Chile

**Keywords:** cryoconcentration, calafate juice, storage time, physicochemical properties, bioactive compounds, antioxidant activity, sensorial analysis

## Abstract

This study aimed to evaluate the potential of centrifugal block cryoconcentration (CBCC) at three cycles applied to fresh calafate juice. The fresh juice and cryoconcentrate at each cycle were stored for five weeks at 4 °C and quality attributes were analyzed every 7 days. CBCC had significant effects in the calafate juice, since in the last cycle, the cryoconcentrate reached a high value of total soluble solids (TSS, ≈42 °Brix), with final attractive color, and an increase of approximately 2.5, 5.2, 5.1, 4.0 and 5.3 times in relation to the fresh juice values, for total bioactive compounds (TBC), 2,2-diphenyl-1-picrylhydrazyl (DPPH), 2,2′-azino-bis(3-ethylbenzothiazoline-6-sulfonic acid) (ABTS), ferric reducing antioxidant power (FRAP) and oxygen radical absorbance capacity (ORAC), respectively. However, at 35 days under storage, these values decreased by 5%, 13%, 15%, 19%, 24% and 27%, for TSS, TBC, DPPH, ABTS, FRAP and ORAC, respectively. Additionally, until the day 14, the panelists indicated a good acceptability of the reconstituted cryoconcentrate. Therefore, CBCC can be considered a novel and viable technology for the preservation of quality attributes from fresh calafate juice with interesting food applications of the cryoconcentrates due to their high stability during storage time in comparison to the fresh juice.

## 1. Introduction

Calafate (*Berberis microphylla*) has presented unique properties due to the high variety of health associated compounds, such as phenols, vitamins, minerals and amino acids [1]. Calafate belongs to the family Berberidaceae, and the fruits are harvested in numerous Chilean and Argentinian Patagonian sectors [2]. However, calafate production is still low compared to other berries (2019: 0.2 hectares calafate versus 970.6, 9.0 and 2.9 hectares of blueberries (*Vaccinium corymbosum*), maqui (*Aristotelia chilensis*) and michay (*Berberis darwinii*), respectively). Therefore, different technologies have been applied to keep this fruit available throughout the year. Thus, processed products from fresh calafate fruits such as juice, jellies, jams and wines can be found in local markets [3].

In recent decades, traditional thermal methods (evaporation, pasteurization and/or sterilization) have been used in fresh juice, and thus, the thermally treated concentrate products achieve a significant improvement in quality and prolonged shelf-life when compared to the fresh juice. Unfortunately, these technologies use high temperatures that cause undesirable changes on different properties (nutrients, flavor and color, among others), since these quality properties contain endless thermolabile and thermostable compounds that are degraded, and thus, the organoleptic properties are affected, resulting in possible rejection by consumers [4]. Hence, emerging non-thermal technologies have been investigated in the food processing sector to concentrate liquids, and thus, to retain their quality attributes [5].

Cryoconcentration (CC) is a non-thermal concentration technology which has demonstrated numerous advantages to preserve important properties in liquid foods [6]. Specifically, CC concentrates a liquid solution by total or partial freezing of water, and thus, as the temperature decreases the solutes are rejected from the ice phase and accumulate at the solid–liquid interphase, i.e., the cryoconcentrated (unfrozen liquid fraction) is between the ice crystals (solid water). Once the freezing process is finished, the cryoconcentrate is removed from the ice fraction, which allows lower energy consumption than thermal processing (0.33 versus 2.26 kJ/g, respectively) [7].

Different CC techniques can be found in the literature, with block CC (BCC) characterized by the easier fraction separation, equipment used and operation procedures [8]. Specifically, in BCC, a liquid solution is completely frozen, which is equivalent to a frozen block solution. Later, the block sample is thawed and separated by a natural (gravitational) method [9] or by employing assisted techniques to improve the process parameters involved in BCC, among them, efficiency, solute yield and percentage of concentrate [10,11,12].

Hence, BCC has proven to be an environmentally friendly emerging technology with great potential to retain various quality characteristics in fresh fruit juice, including physicochemical parameters [13], phenolic content [14], antioxidant activity [15] and volatile compounds [16]. In addition, sensory panelists did not find differences between reconstituted cryoconcentrate samples and fresh juice [17]. Nevertheless, to our knowledge, no researches have been published on quality characteristics obtained from a fresh native juice such as calafate juice by BCC. Therefore, the novelty of this study is the concentration of calafate juice, an endemic species of the Patagonian Andes of Chile and Argentina considered as a “superfruit” with high polyphenol content and high antioxidant capacity through a green and non-thermal technology called cryoconcentration.

Therefore, the aim of the study was to investigate the stability of fresh calafate juice and cryoconcentrate samples, in quality attributes terms, during 35 days of storage after applied centrifugal BCC (CBCC) technology. The physicochemical parameters, bioactive compounds content, antioxidant activity and sensory analysis were studied in each week of the storage period.

## 2. Materials and Methods

### 2.1. Calafate Juice Preparation

Fresh calafate (*Berberis microphylla*) were harvested in southernmost Chile (XI Región de Aysén) (December 2019), and the fruits were transferred in a refrigerated truck to Chillán (Región del Ñuble, Chile). The fruits were pressed, and then, the juice was filtered through nylon cloth (0.8 mm fine-mesh) to discard solid parts (peel and seeds) that might interfere with the CBCC process. The liquid sample was stored at 4 °C and processed within 24 h.

### 2.2. CBCC Protocol

The CBCC process was carried out as described in our previous study [18]. Thus, the prepared calafate juice (45 mL) was placed in plastic centrifugal tubes with foamed polystyrene (around the tube) to produce an axial freezing. The samples were frozen at −20 °C (overnight) in a vertical static freezer (280, M and S Consul, Sao Paulo, Brazil), and at the end of the freezing stage, the frozen samples were transported to centrifuge equipment (Eppendorf 5430R, Hamburg, Germany). Specifically, two centrifugation conditions (15 min with 4000 rpm and 20 min with 4000 rpm) were performed to determine the best separation condition, considering total soluble solids (TSS), efficiency (Eff, %) and final cryoconcentrate volume (mL). Thereby, the centrifugation was used as an assisted technique to force the extraction of the cryoconcentrated fraction (C_s_) from the frozen matrix (C_f_). The CBCC was performed at three cycles, i.e., the C_s_ at the first cycle was used as feed solution for the second cycle and the second C_s_ was used for the third cycle.

### 2.3. Physicochemical Analysis

The TSS was determined using a digital refractometer PAL-3 (range: 0–93 °Brix, precision: ±0.1 °Brix, Atago Inc., Tokyo, Japan). The density (kg/m^3^) of the samples was determined by the pycnometric method at 20 °C using distilled water as a model liquid [19]. A digital pH meter HI 2221 (Hanna Instruments, Woonsocket, RI, USA) was used to determine the pH of the samples, and the mean pH values were calculated on the International Union of Pure and Applied Chemistry (IUPAC) recommendation [20]. The titratable acidity (TA) was measured by using 5 mL of sample mixed with 50 mL of degassed deionized water, pH of samples was adjusted to 8.2 with sodium hydroxide solution (0.1 M NaOH) and the TA was expressed as grams of malic acid (MA) per liter of sample (g MA/L). The color parameters were calculated on the International Commission on Illumination (CIE) with L* (Lightness), a* (Green–red axis) and b* (Blue–yellow axis) space (CIELAB) using a spectrophotometer CM-5 (Konica Minolta, Osaka, Japan). The standard illuminant and observer angle were D65 and 10°, respectively. In addition, the total color difference (ΔE*) between fresh calafate juice and cryoconcentrated samples was calculated according to Equation (1).
(1)$$\Delta E^{*}=\sqrt{{\left(\Delta L^{*}\right)}^{2}+\;{\left(\Delta a^{*}\right)}^{2}+{\left(\Delta b^{*}\right)}^{2}}$$
where ΔL*, Δa* and Δb* are differences between fresh calafate juice and cryoconcentrated samples at each week of the storage period.

TSS, pH, acidity, density and color determinations of fresh juice and cryoconcentrated samples were performed in triplicate at ambient temperature (≈22 °C). Three replicates for each treatment were analyzed.

### 2.4. Quantification of Total Bioactive Compound (TBC)

The total polyphenol content (TPC), total anthocyanin content (TAC) and total flavonoid content (TFC) of fresh calafate juice and cryoconcentrated samples were measured at each CBCC cycle and each storage period.

TPC was determined through the Folin–Ciocalteau method [21]. Wherein, 200 µL of sample and 1500 µL of diluted (1:10) Folin–Ciocalteau reagent were mixed. After 5 min, 1500 µL of sodium carbonate solution (20% (*w/v*), Na_2_CO_3_) was added to the solution. After 90 min in the dark at room temperature (incubation), the absorbance was measured at 760 nm. Gallic acid (GA) was used for the standard curve construction, and the TPC results were expressed as mg of gallic acid equivalents (GAE) per grams (g) of dry matter (mg GAE/g d.m.).

TAC was quantified using the pH differential method [22]. Therefore, 200 µL of sample was added to 1800 µL of potassium chloride (pH 1.0, 0.025 M, KCl) and 1800 µL of sodium acetate (pH 4.5, 0.4 M, CH_3_COONa). After 30 min in the dark at room temperature (incubation), the absorbance was measured at 520 and 700 nm. Cyanidin-3-glucoside (C3G) was used for the standard curve construction, and the TAC results were expressed as mg of C3G equivalent per grams (g) of dry matter (mg C3G/g d.m.).

TFC was measured by the aluminum chloride colorimetric method [23]. As such, 250 µL of sample was mixed with 1000 µL of distilled water and 75 µL of sodium nitrite solution (5% (*w/v*), NaNO_2_). After 10 min, 75 µL of aluminum chloride (10% (*w/v*), AlCl_3_), 500 µL of sodium hydroxide (1 M, NaOH) and 600 µL of distilled water were added. After 30 min in the dark at room temperature (incubation), the absorbance was measured at 510 nm. Catequin (C) was used for the standard curve construction, and the results were expressed as mg of catechin equivalent (CE) per grams (g) of dry matter (mg CE/g d.m.).

All TBC determinations were evaluated using a spectrophotometer T70 UV–VIS (Oasis Scientific Inc., Greenville, SC, USA), and were done in triplicate at ≈22 °C.

### 2.5. Total Antioxidant Activity (TAA) Determinations

Four methods were used to quantify antioxidant activity, the 2,2-diphenyl-1-picrylhydrazyl (DPPH), 2,2′-azino-bis(3-ethylbenzothiazoline-6-sulphonic acid) (ABTS), ferric reducing antioxidant power (FRAP) and oxygen radical absorbance capacity (ORAC) assays, with minor modifications.

DPPH assay was determined using the protocol described by Brand–Williams et al. [24]. Thereby, 100 µL of sample was added to 2900 µL of DPPH solution (0.1 mM). The solution was incubated in the dark at room temperature (≈22 °C) for 30 min, and then, the absorbance was measured at 515 nm.

ABTS assay was performed according to Re et al. [25]. Therein, 10 μL of sample was added to 990 μL of ABTS solution. The sample was incubated in the dark at room temperature for 30 min, and then, the absorbance was measured at 734 nm.

FRAP assay was done according to the method described by Benzie and Strain [26]. As such, 100 µL of sample, 3000 µL of FRAP reagent and 300 μL of water were mixed. The sample was incubated in the dark at 37 °C for 10 min, and then, the absorbance was measured at 593 nm.

DPPH, ABTS and FRAP assays were quantified on a spectrophotometer T70 (UV–VIS spectrophotometer, Oasis Scientific Inc., Greenville, SC, USA).

ORAC assay was determined using the method reported by Ou et al. [27]. Specifically, 30 μL of sample and 20 μL of fluorescein solution (10 nM) were placed into black 96-well microplates and incubated at 37 °C for 30 min. Then, 50 μL of 2,2′-azobis(2-amidinopropane) dihydrochloride (AAPH, 600 mM) and 2900 μL of phosphate buffer (75 mM, pH 7.4) were added to the solution. The absorbance was measured every 1 min for 60 min at an excitation wavelength of 485 nm and an emission set of 520 nm using a multimode plate reader (Victor X3, Perkin Elmer, Hamburg, Germany).

For all assays, Trolox (T) (6-hydroxy-2,5,7,8-tetramethylchroman-2-carboxylic acid) was used for the standard curve construction, and the TAA results were expressed as mM Trolox equivalents (TE) per gram (g) of dry matter (mM TE/g d.m.) and the TAA determinations were performed in triplicate.

### 2.6. Storage Study

The fresh calafate juice and each cryoconcentrate sample were deposited in glass jars previously rinsed with distilled water and UV exposed for 1.5 h. All the samples were stored at 4 ± 1 °C in a refrigerated incubator (FOC 215E, Velp Scientific Inc., Milano, Italy) for 35 days. The physicochemical properties, TBC and TAA determinations were analyzed at day 0 (control) and after 7, 14, 21, 28 and 35 days, as previously described.

### 2.7. Sensory Evaluation

A sensorial analysis was done to measure the degree of acceptance or rejection between reconstituted cryoconcentrated samples (third cycle with similar TSS value than the fresh juice) and fresh calafate juice. The evaluations were performed at day 0, 7, 14 and 21 of the storage period by a trained sensory panel consisting of ten males and ten females with an average age of 33 years old (from 27 to 39 years old). The samples were rated according to a 5-score hedonic scale system, in which 1 = dislike extremely and 5 = like extremely. Thus, odor, aroma, flavor and overall acceptability were evaluated. Specifically, 20 mL of samples at 22 °C were placed in transparent cups, labeled with three random numbers. Cold water and crackers were supplied to each panelist in each test for rinsing their mouths between the samples. Three replicates were performed on each sample.

### 2.8. Statistical Analysis

The results were expressed as means ± standard deviation. All statistical analysis was evaluated by analysis of variance (ANOVA) test and the treatment means were compared via least significant difference (LSD) or Student’s t-test at level of significance (*p* ≤ 0.05). Statgraphics Centurion XVI software version 16.2.04 (StatPoint Technologies Inc. Warrenton, VA, USA) was used for analysis of data. Correlations between TBC, TAA and among them were evaluated by Pearson’s correlation coefficient test.

## 3. Results and Discussion

### 3.1. Preliminary Centrifugation Results

In order to determine an adequate CBCC process, two centrifugation time conditions (15 min and 20 min) were used to identify the best separation performance. The other conditions were similar to that previously reported in our laboratory, i.e., 4000 rpm and 20 °C as centrifugation speed and separation temperature, respectively [16,17,18].

The centrifugation time had an important effect in TSS, efficiency and final cryoconcentrate volume, with significant differences (*p* ≤ 0.05) at each cycle (Table 1). Firstly, a gradual increase in TSS values was observed as cycles progressed, in both 15 min and 20 min conditions. However, at 15 min of centrifugation, the TSS values (27, 38 and 45 °Brix) were higher than those at 20 min (24, 31 and 37 °Brix), in the first, second and the third cycle, respectively. These results could be attributed to the centrifugation which remained relatively intact the ice fraction (without thawing and/or breaking) using 15 min as centrifugation time, and thus, only cryoconcentrate was extracted from the ice fraction.

An inverse behavior was observed in efficiency, since 20 min presented better results than 15 min, with 63% and 52%, in the last cycle, respectively. This phenomenon can be described by the TSS concentration values in the C_s_ and C_f_ fractions in each cycle, since a high TSS value produces an increase in viscosity and thus, an increase the difficulty of solute extraction from the ice matrix, which produces a reduction in the separation efficiency [28].

An important point to determine the best centrifugation condition is the cryoconcentrate volume (CV), which decreased as the cycles advanced. In our case, centrifugation for 20 min presented higher CV post-centrifugation by tube than 15 min, from ≈34 to ≈22 mL and ≈29 to ≈15 mL, respectively. These results can be correlated with the previously results obtained in TSS, since, as mentioned above, a high TSS value increases the viscosity, and this behavior influences the CV obtained due to the high viscosity preventing the solutes movement outside from the frozen phase.

Therefore, we defined 20 min as centrifugation time for the separation between C_s_ and C_f_ at each cycle, since it allows a higher CV extraction than 15 min. In addition, calafate (*Berberis microphylla*) is still a fruit with low production in Chile compared to other berries [29], and to date, these fruits present a high price per kilogram due to recent knowledge acquired on different quality properties [1,2,3], which encourages the cryoconcentrated juice study and stability over time for the elaboration of different products such as fruit juice.

### 3.2. TSS Results

The fresh juice has an initial TSS value close to 13.9 °Brix (Figure 1), which is a value within the range established by Mariangel et al. [3], who studied the variability in numerous attributes of calafate fruit harvested from four sectors in Southern Chile, with TSS values between 9.3 to 22.9 °Brix, which reflects the influence of different agricultural and climatic conditions on the fruit growth pattern.

Firstly, in relation to day 0, the TSS in the fresh juice increased significantly cycle to cycle with values of 23.1, 35.6 and 42.0 °Brix in the first, second and third cycle, which is equivalent to a concentration index (CI, ratio C_s_/C_0_) of 1.7, 2.6 and 3.0 times, compared to the initial TSS value (13.9 °Brix), respectively. Thus, the TSS results at each cycle have higher values than those achieved in previous investigations with other fresh fruit juice samples such as blueberry juice [18,30], orange juice [10] and pineapple juice [16] under similar conditions in our laboratory, with final concentration values (third cycle) close to 41, 33, 40 and 36 °Brix, respectively. Furthermore, the TSS values were superior to those described by Moreno et al. [31] and Ding et al. [32], who used BCC and suspension CC (SCC) to cryoconcentrate coffee extract and apple juice, respectively. Hence, the TSS values variation could be explained by the freezing conditions and sized tubes capacity used in the present study. Specifically, we used an axial freezing front propagation with moderate freezing rate temperature (−20 °C), which allows an improved counter-diffusion of solutes from the growing crystal surface. Additionally, the centrifugal equipment has a sized tubes capacity of 50 mL-tube, which favors the cryoconcentrate extraction from the ice frozen in the centrifugation step [18].

During the first week (day 7) under storage, the TSS in the fresh juice presented a slight increase (with statistical differences) compared with control juice (day 0). However, a continuous decline in TSS was observed in the next weeks, with a final value of approximately 6.0 °Brix at day 35 (week 5), which is equivalent to a decrease of more than 57% of the initial TSS value (day 0). A similar effect was observed for all CBCC cycles, with a slight increase until day 7, and then, TSS decreased significantly until day 35, where it reached values close to 17.8, 31.8 and 40.0 °Brix, for the first, second and third cycle, which indicates a decrease of 23%, 11% and 5%, with respect to the correspondent value at day 0, respectively.

The TSS decrease at each cycle under storage could be associated with sugar consumption by microorganisms, since as time progresses, there exists a possibility of microbial growth, and thus, as days passed, the microorganisms consume higher sugar amount than in the first days (first week), reflecting in a gradual TSS decrease [33]. Comparable results were described by Wahia et al. [34], who studied melon juice and their quality properties preservation at various days during storage. Additionally, Chia et al. [35] mentions that, in terms of consumption safety, the unpasteurized products has a shelf life up to two weeks, since, in general, the fresh fruits have a microbial load close to 3 to 5 log CFU per mL and the limit is 6 log CFU per mL.

### 3.3. Physicochemical Analysis

Statistical differences (*p* ≤ 0.05) were found in density, pH and acidity values between the fresh juice and their respective CBCC cycle, and in turn, significant differences (*p* ≤ 0.05) were observed between the samples (at day 0) and their correspondent at each week under storage (Table 2).

Firstly, the initial pH and TA values in the calafate juice (day 0) were equivalent to those previously reported by Arena et al. [36], who studied the organic acids content of calafate in different growing seasons, specifying that the physicochemical properties of calafate depends on various aspects such as climate, place of growth, type of harvest and processing procedures.

Specifically, in the fresh juice (day 0) and as the cycles advanced, a gradual decrease in the pH values were observed, with a decrease close to 2.3%, 5.5% and 8.7% in relation to the initial pH value (pH ≈3.09), for the cycle one, cycle two and cycle three, respectively. While, an opposite effect was denoted in TA values, since it presented a considerable increase, with values of approximately 2.1, 2.8, 4.1 and 4.6, from fresh juice to the final CBCC cycle, respectively. This contrary performance has been linked to the TSS and their values cycle by cycle, i.e., as TSS increased, an increase in the organic acid content was generated, affecting the pH and TA values [37]. Besides, the results are in accordance with those obtained in CC applied to pineapple juice [16], apple juice [17] and blueberry juice [38], in which all the cryoconcentrated juices had antagonistic values in pH and TA with the increase in solutes as the cycles passed.

On the days under storage, an opposite behavior was observed in each sample among pH and TA values, since a progressive increase was detected in pH and a significant decline was identified in TA, with values from 3.1 to 3.4 and 2.1 to 1.7 for fresh juice, 3.0 to 3.3 and 2.8 to 2.3 for cycle one, 2.9 to 3.2 and 4.1 to 3.6 for cycle two and 2.8 to 3.2 and 4.6 to 4.0 for cycle three, from the day 0 to day 35, respectively. In this case, this phenomenon has been attributed to the acid hydrolysis of various polysaccharides, in which the non-reducing sugars are transformed into reducing sugars, as well as the use of malic acid as an energy source by microorganisms [39].

In general terms, the fruit juices are a good media for microbial multiplication and spoilage, with a usual microbial increase during storage time. In these conditions, the microorganisms use nutrients and cause enzymatic changes, contributing to creating off-flavor by breakdown or synthesis of new compounds [40]. In the case of unpasteurized fruit juices, the microbial spoilage is most commonly the result of aciduric microbes such as lactic acid bacteria and yeasts that produce copious quantities of carbon dioxide and off-flavors [41].

Similar trends were perceived in fruit juices such as sugarcane juice [42], grape juice [43] and apple juice [44] during storage.

In terms of density, the values showed an overall increasing trend from the fresh juice to the last cycle at day 0, and thus, the density values presented an increase close to 5%, 8% and 12% in comparison to the respective initial value. This behavior can be justified by the TSS concentration reached post-centrifugation step at each cycle [45] and the performance was comparable with the results informed for different cryoconcentrated juices [16,17,38,45]. In addition, a similar behavior was observed throughout storage, since at day 35, the density showed an increment of approximately 12–14% to the respective sample at day 0. These trends might be explained by the water evaporation in the sample under storage period, which lead to a decrease in the volume of the sample, as it was explained in studies on physicochemical properties under storage conditions for watermelon juice [33] and grape juice [46].

Color parameters of fresh calafate juice and for each CBCC cycle during storage are shown in Table 3. The samples exhibited significant changes at each cycle, mentioning that the differences among the samples were quantitative and qualitative.

At day 0, the fresh calafate juice presented a light reddish violet color (L* ≈ 34.0, a* ≈ 31.2 and b* ≈ 3.1). However, to date, there are no studies that report the CIELAB color coordinates for calafate juice, but the color can be related to other fruit juices with comparable tonality such as strawberry juice [15], blueberry juice [18] and pomegranate juice [47]. Thus, at each CBCC cycle, a remarkable modification with respect to the fresh juice was observed, since L* and b* values decreased, i.e., the concentrate samples were darker with a slight brown tone, while a* values had a significant increase, demonstrating a trend from the light red to dark red color (Figure 2). This behavior could be accredited to the increase in TSS values, since as the cycles advanced; more concentrated solute was separated from the ice fraction, leading to an intensification of the natural fruit juice color. These CIELAB values presented concordance with previous results for cryoconcentrated orange juice [10], strawberry juice [15], blueberry juice [18] and apple juice [48,49], in which the juices were darker with marked increase in a* and/or b* coordinates, depending on the initial juice color, as the cycles increased.

In storage terms, at day 35, the fresh juice offered a considerable decrease in the L* and b* values by 60% and 36%, and a* values had an increase by 34% with respect to the initial CIELAB values (day 0), respectively. Similarly, in the last cycle, a pronounced decrease by 93% and 97% was observed in L* and b* values in comparison to the values in the same cycle at day 0 (L* = 7.22 and b* = 1.21), respectively. On the contrary, a* indicates an increase by 18% with respect to the third cycle at day 0 (a* = 46.71). Comparable performances, under different days of storage, were reported by Igual et al. [46] for grape juice, Yildiz and Aadil [50] for strawberry juice and Wurlitzer [51] for tropical fruit juices, specifying that the darkening and tendency to brown color is due to the compounds degradation by factors such as nonenzymatic Maillard reaction, exposure to air and light, pH changes and enzymatic activities, which leads to the oxidation in the sample that alters the visual appearance of the juice. Furthermore, the visual color of fresh calafate juice and each cycle during storage are presented in the Appendix A, in which it is possible to observe the change from light reddish violet (fresh juice, day 0) to an attractive dark reddishness color due to the components concentration in the fresh calafate juice, as the cycles advanced. However, as the days passed, the color had a darker tone, which turned a brown color (cycle two) and a dark brown color (cycle three), which displays the degradation throughout the storage period, as mentioned above.

According to ΔE* evaluation (Table 3), at day 0, the values were over 14 CIELAB units, indicating that the human eye can find differences between fresh juice with each CBCC cycle, based on the scale proposed by Pankaj et al. [52] (ΔE* ≥ 3, the color is humanly perceptible). The difference was more pronounced between fresh juice and the third cycle, since the ΔE* value was close to 31 units, indicating that the tendency to a dark reddish color generates a significant visual change to the fresh juice. As storage time passed, the ΔE* values were less noticeable than the samples at day 0. These ΔE* values were clearly depending on the TSS in the sample, since a high ΔE* between the fresh juice and cryoconcentrates was obtained by increasing the TSS cycle to cycle, and in turn, it can be related with the change in L* values. As an important point, the decrease in the ΔE* values under storage time in comparison at day 0 is due to the progressive component degradation as days passed, with the color going from a light reddish violet to a red color, and from dark red to a dark brown, for the fresh juice and third cycle, from the day 0 to day 35, respectively, reducing the visual perception when the samples were compared.

### 3.4. TBC of Fresh and Cryoconcentrated Samples

TBC values at day 0 and during storage time of fresh juice and cryoconcentrated samples are presented in Table 4.

The TBC values (TPC, TAC and TFC) in fresh calafate juice (day 0) were close to 54.7 mg GAE/g d.m., 41.2 mg C3G/g d.m. and 31.9 mg CE/g d.m., respectively. These results were superior to those found by Brito et al. [53], who studied various bioactive components of native berries from VIII Region of Chile. The variations in TBC values could be due to the geographical characteristics in each Region, since there is a distance of approximately 1200 km between VIII Region and XI Region, which leads to diverse environmental conditions, genetic and species variabilities, affecting the time and form of fruit maturity and harvest-type method used by farmers, and thus, all these factors could explain the disparity between TBC values.

At day 0, the TBC values significantly increased as the cycles progressed, with an increase of approximately 1.2, 2.0 and 2.7, 1.1, 1.9 and 2.5 and 1.1, 1.8 and 2.4 times in relation to the initial TBC value, for TPC, TAC and TFC, for cycle 1, cycle 2 and cycle 3, respectively. Various studies showed a similar upward trend for TBC in the cryconcentration of orange juice [10], maqui juice [14], strawberry juice [15], pineapple juice [16], apple juice [17,32], blueberry juice [38] and broccoli extract [54], i.e., since as liquid food was concentrated at low temperatures, the thermolabile components were highly protected in comparison to high thermal concentration technology as evaporation, which seriously affects the bioactive composition of the liquid food due to the high processing temperatures [6].

In the first week of storage, all the samples presented a significant increase in TBC values, which exceeded the initial TBC value between 4% to 11%. This effect could be associated to the TSS behavior (Figure 1), since in the same week (day 7), the TSS values were higher than the initial results, and later, it decreased considerably during the weeks. Therefore, TSS and TBC are directly proportional. A clear and drastic decrease in TBC values was observed in the next weeks, and at the end of 35 days of storage, there was a reduction of up to 22%, 16%, 7% and 4% for TPC, 31%, 25%, 14% and 8% for TAC and 44%, 41%, 31% and 25% for TFC, with respect to the initial TBC value (day 0) for fresh juice, cycle one, cycle two and cycle three, respectively. This decline in the TBC values can be linked by factors as oxidation and/or polymerization of phenolic compounds with various proteins, and condensation of pigments with phenolic compounds present in the juice. Besides, the TBC degradation during storage time has been related to peroxidase enzyme activity [55]. However, to date, there are no studies on CC and enzymes that degrade bioactive components such as peroxidase, allowing the opening to future research.

### 3.5. TAA of Fresh and Cryoconcentrated Samples

The TAA values (mM TE in 100 g, on dry matter) in the fresh juice were approximately 6.9, 14.7, 23.0 and 21.4 for DPPH, ABTS, FRAP and ORAC, respectively (Table 5). These values are in line with values reported by Ruiz et al. [1], Mariangel et al. [3] and Brito et al. [53], who studied the antioxidant activity of calafate from different harvest seasons and geographical areas in Southern Chile, indicating that the variation in TAA values could be due to specific climatic and agricultural environments, since each Region presents endless characteristics that affect the genetic and growth of the fruits, pre-harvest phases, ripening, post-harvest processing, among other conditions, and thus, all these conditions impacts on the composition of the fruit.

As in TSS and TPC, as the cycles advanced, the differences in TAA values were statistically significant between the fresh juice and each cycle, with an increase of 2.5, 3.9 and 5.2-fold, 2.6, 3.9 and 5.1-fold, 1.9, 2.9 and 4.0-fold and 3.0, 4.1 and 5.3-fold, in comparison to the initial TAA values (fresh juice, day 0), for cycle one, cycle two and cycle three, for DPPH, ABTS, FRAP and ORAC, respectively. This upward behavior indicates a direct relationship with TSS and TBC values, since both values increased as the cycles progressed. The results are in agreement with CC reports applied to different liquid foods [15,48,56]. Hence, our results confirm that the conditions used to concentrate calafate juice allows an increase of TAA values, and thus, the sensitive components, as total anthocyanin content, are preserved, and these contribute to a high TAA [16,17,57,58].

As weeks passed, the TAA was progressively degraded, and at day 35, the fresh calafate juice was the sample most affected by storage time, with a decrease close to 40%, 42%, 48% and 50%, while, the third cycle gave lower TAA losses than the other samples, with reduction of approximately 15%, 19%, 24% and 27% in relation to the fresh juice value (day 0), for DPPH, ABTS, FRAP and ORAC, respectively. The downward trend of TAA values during the storage for several weeks are in agreement with other studies on fruit juices, which have described continuous TAA degradation such as sugarcane juice [42], grape juice [43,46], strawberry juice [50,59] and orange juice [60], which indicate that the oxidation of bioactive compounds and polymerization reactions of anthocyanins could be linked to the loss of antioxidant activity as the days passed.

### 3.6. Correlation between TBC and TAA

Correlation coefficients between TBC and TAA for cycle three are shown in Table 6. Furthermore, the correlation coefficients for fruit juice, cycle one and cycle two are presented in the Appendix A.

In all the samples, a positive and significant correlation was found between the biological active compounds content. The values (r) between 0.9 to 1.0, indicating the direct and proportional results between TBC and TAA, i.e., each component increased or decreased, as the cycles or the days under storage progressed, respectively. A similar trend was distinguished by Casas–Forero et al. [38] and Correa et al. [57], who reported a high correlation (0.9 to 1.0) between antioxidant activity and bioactive compound in CC applied to blueberry juice and aqueous coffee extract, respectively.

### 3.7. Sensorial Analysis

The acceptance scores attributed by the panelists between fresh calafate juice and reconstituted cryoconcentrated juice (third cycle) are represented in Figure 3.

In relation to the day 0, there were no significant differences in odor, aroma, flavor and global assessment between the fresh juice and reconstituted cryoconcentrated sample. Specifically, all the values were assessed as “like”, since the evaluations were superior to four points, i.e., the panelists specified a pleasure when tasting the fresh and cryoconcentrated juices, without finding differences when comparing the quality sensorial characteristics.

At day 7, the hedonic scale scores decreased significantly compared at day 0, with values between 3.0 and 3.8 points, which is equivalent to the category “liked slightly”. Therefore, the samples were considered as accepted for the sensorial panelists. Despite the decline, there were no statistical differences between the scores assigned by the panelists for the sensory attributes among the samples.

Nonetheless, as days advanced, the panelists reported an increase in the degree of rejection between samples, since the scores decreased considerably, and in addition, the points between the fresh juice and cryoconcentrated varied significantly. Specifically, the cryoconcentrated had better acceptance than the fresh juice, with scores of approximately 2.8–3.0 versus 2.4–2.9 and 2.1–2.5 versus 1.1–2.0, for day 14 and day 21, respectively. Thus, these scores designated a negative impression of the samples with respect to their sensorial characteristics. However, from 14 days in storage, the cryoconcentrates were more accepted to the sensory panel than the fresh juice, reinforcing that CC maintains better sensory attributes in the reconstituted concentrates than fresh juice during refrigerated storage.

Based on these results, at day 14 and day 21, the cryoconcentrated samples presented better scores than the fresh juice, since, as mentioned above, previous studies have shown that CC technology allowed the obtaining of high TSS values, and in turn, these increases the volatile compounds release [17]. Therefore, the reconstituted juice contains a high organoleptic acceptance that can be connected to taste and aroma, and thus, it resist storage time better than fresh juice, since the fresh juice possibly had a high fermentation level, generating more rancid odors and flavors on day 21 [61], i.e., the fresh juice was more vulnerable to external factors [62], which eventually led to a high degree of rejection. However, to date, there are no studies that compare a fruit juice with a reconstituted cryoconcentrated juice in terms of bioactive components, antioxidant capacity, volatile compounds and physicochemical properties. Therefore, a very interesting study about the commercialization of cryoconcentrated juice could be realized with similar concentration that a fresh juice.

Sensory analysis results on days 28 and 35 were not presented, since on day 21, the panelists indicated a high disgust and complete rejection, with values less than 2.5 point for all the characteristics evaluated.

Therefore, CC technology allows the preservation of different sensory attributes, and in turn, its concentration at low temperatures proves a high similarity in the reconstituted concentrate to the original sample. Analogous trends were observed in CC studies applied to apple juice [17], coffee extract [31], black currant juice [63] and Andes berry pulp [64].

## 4. Conclusions

CBCC positively affects the quality properties of fresh calafate juice, as exemplified by the high TSS values obtained in the last cycle (42 °Brix). Additionally, this non-thermal technology intensifies the natural juice color to an attractive dark reddish color. Subjecting fresh calafate juice to CBCC resulted in high TBC content, with values over 2.4 times the initial TBC values. Moreover, the TAA presented a similar behavior, since these increased between 4.0 to 5.3 times, in comparison to the initial TAA values. These values indicate the advantages of CBCC application as a green technology to extract a high amount of concentrated liquid from the ice matrix without the use of high temperatures.

Under refrigerated storage time of 35 days, the CBCC samples showed better stability than the fresh juice, since the cryoconcentrated samples showed a low decreasing rate in their nutritional properties, inferring that CBCC allows to concentrate and to retain various bioactive and antioxidant components naturally present in calafate juice.

Additionally, until the day 21, sensory panelists reported acceptability of the reconstituted cryoconcentrated sample, while the fresh juice was totally rejected, reinforcing that CBCC also concentrate and preserve volatile components that are perceived by the panelists as an important quality parameter.

Therefore, CBCC can be considered a novel and viable technology for the preservation of quality attributes from fresh native juice with interesting food application of the cryoconcentrates due to its high stability during storage time and low bioactive components degradation in comparison to the fresh juice.

## Figures and Tables

**Figure 1 foods-09-01314-f001:**
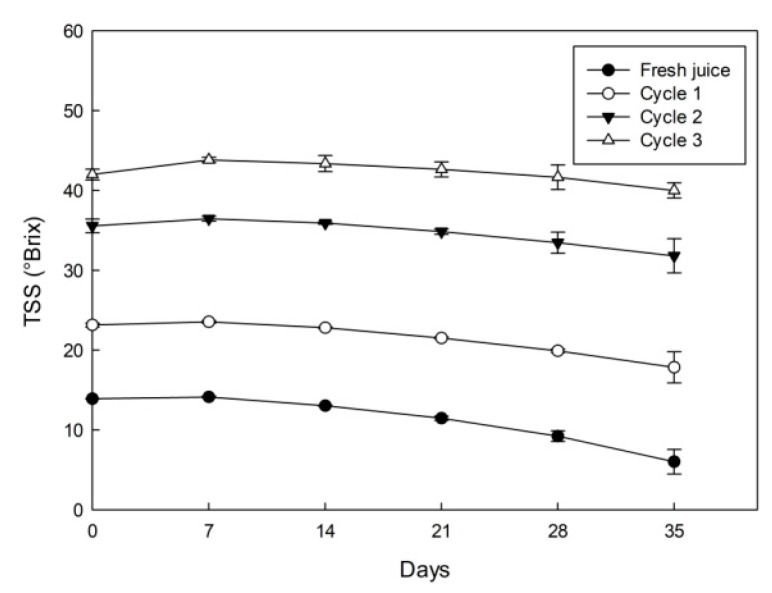
Effect of storage on TSS in fresh and cryoconcentrated calafate juice.

**Figure 2 foods-09-01314-f002:**
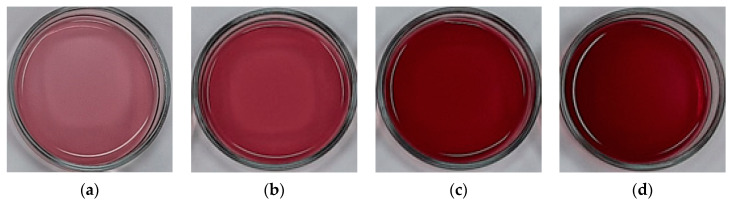
Visual appearance of samples: (**a**) fresh calafate juice; (**b**) cycle 1; (**c**) cycle 2; and (**d**) cycle 3.

**Figure 3 foods-09-01314-f003:**
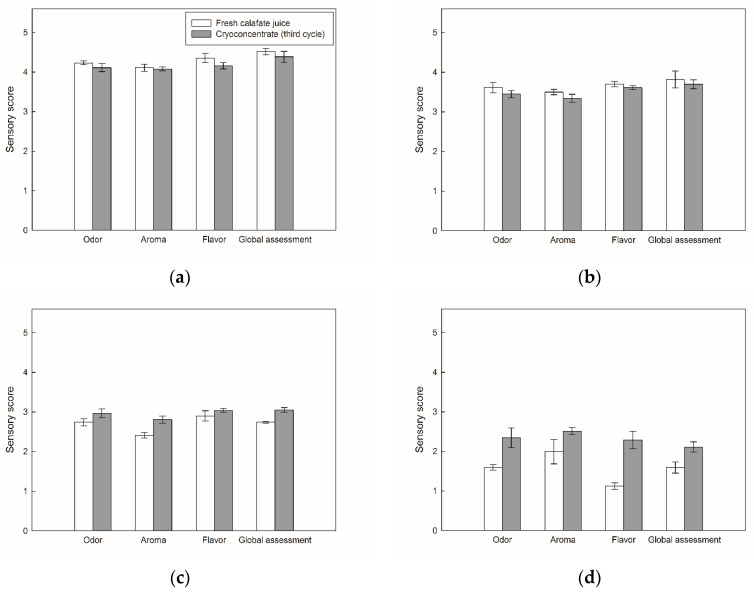
Sensory attributes between fresh calafate juice and reconstituted cryoconcentrated sample at the third cycle: (**a**) day 0; (**b**) day 7; (**c**) day 14; and (**d**) day 21.

**Table 1 foods-09-01314-t001:** Preliminary results obtained at different centrifugation conditions.

Centrifugation	Cycle	TSS	Efficiency ^1^	Cryoconcentrated Volume
Condition		(°Brix)	(Eff, %)	(CV, mL)
Fresh juice	0	14.6 ± 0.2 ^a^	-	-
	1	27.2 ± 0.8 ^b^	68.9 ± 0.9 ^a^	28.6 ± 0.2 ^a^
15 min, 4000 rpm, 20 °C	2	37.6 ± 0.4 ^c^	57.9 ± 0.4 ^b^	22.4 ± 0.6 ^b^
	3	44.5 ± 0.5 ^d^	52.3 ± 1.1 ^c^	15.2 ± 0.8 ^c^
	1	23.7 ± 0.7 ^b^	75.7 ± 1.0 ^a^	34.4 ± 0.6 ^a^
20 min, 4000 rpm, 20 °C	2	31.1 ± 0.6 ^c^	69.7 ± 0.9 ^b^	29.1 ± 1.0 ^b^
	3	36.6 ± 0.3 ^d^	63.0 ± 0.5 ^c^	21.6 ± 0.7 ^c^

Total soluble solids (TSS), efficiency (Eff) and cryoconcentrate volume (CV). Different superscripts in a column are significantly different (*p* ≤ 0.05) according to least significant difference (LSD) test. ^1^ Eff(%) = ((C_cs_ − C_cf_)/C_cs_) * 100, where C_cs_ and C_cf_ are the solutes in cryoconcentrated and ice fraction, respectively.

**Table 2 foods-09-01314-t002:** Physicochemical parameters of fresh calafate juice and cryoconcentrate samples during storage.

Day	pH				TA (g MA/L)				Density (kg/m^3^)			
	Fresh Juice	C1	C2	C3	Fresh Juice	C1	C2	C3	Fresh Juice	C1	C2	C3
0	3.09 ± 0.01 ^a,A^	3.02 ± 0.01 ^b,A^	2.92 ± 0.02 ^c,A^	2.82 ± 0.02 ^d,A^	2.07 ± 0.01 ^a,A^	2.77 ± 0.05 ^b,A^	4.10 ± 0.09 ^c,A^	4.59 ± 0.06 ^d,A^	1030 ± 3.55 ^a,A^	1080 ± 5.92 ^b,A^	1110 ± 9.01 ^c,A^	1150 ± 3.77 ^d,A^
7	3.15 ± 0.02 ^a,B^	3.07 ± 0.02 ^b,B^	3.00 ± 0.03 ^c,B^	2.89 ± 0.00 ^d,B^	2.03 ± 0.02 ^a,B^	2.65 ± 0.02 ^b,B^	3.97 ± 0.02 ^c,B^	4.40 ± 0.08 ^d,B^	1040 ± 2.74 ^a,B^	1100 ± 10.14 ^b,B^	1140 ± 5.74 ^c,B^	1160 ± 4.58 ^d,B^
14	3.24 ± 0.01 ^a,C^	3.18 ± 0.00 ^b,C^	3.05 ± 0.01 ^c,C^	3.01 ± 0.01 ^d,C^	1.98 ± 0.00 ^a,C^	2.59 ± 0.01 ^b,C^	3.90 ± 0.01 ^c,C^	4.25 ± 0.05 ^d,C^	1060 ± 3.01 ^a,C^	1130 ± 6.63 ^b,C^	1160 ± 2.10 ^c,C^	1200 ± 10.15 ^d,C^
21	3.30 ± 0.03 ^a,D^	3.21 ± 0.01 ^b,D^	3.09 ± 0.02 ^c,D^	3.06 ± 0.00 ^d,D^	1.84 ± 0.02 ^a,D^	2.44 ± 0.00 ^b,D^	3.82 ± 0.03 ^c,D^	4.15 ± 0.03 ^d,D^	1090 ± 10.75 ^a,D^	1150 ± 11.25 ^b,D^	1180 ± 11.52 ^c,D^	1240 ± 12.00 ^d,D^
28	3.36 ± 0.01 ^a,E^	3.25 ± 0.02 ^b,E^	3.17 ± 0.03 ^c,E^	3.11 ± 0.01 ^d,E^	1.79 ± 0.01 ^a,E^	2.37 ± 0.01 ^b,E^	3.75 ± 0.02 ^c,E^	4.10 ± 0.01 ^d,D^	1120 ± 9.95 ^a,E^	1170 ± 4.12 ^b,E^	1210 ± 3.58 ^c,E^	1270 ± 6.83 ^d,E^
35	3.40 ± 0.02 ^a,F^	3.29 ± 0.01 ^b,F^	3.22 ± 0.01 ^c,F^	3.17 ± 0.02 ^d,F^	1.72 ± 0.03 ^a,F^	2.32 ± 0.01 ^b,F^	3.62 ± 0.04 ^c,F^	4.03 ± 0.04 ^d,E^	1150 ± 12.10 ^a,F^	1200 ± 6.58 ^b,F^	1250 ± 4.50 ^c,F^	1310 ± 4.45 ^d,F^

a–d: Different small letters in the superscript in the same row denote differences at 5% between the fresh calafate juice and their cycles, according to the LSD test. A–F: Different capital letters in the superscript in the same column denote differences at 5% in the sample during storage time, according to the LSD test. C1, C2 and C3 represents cycle 1, cycle 2 and cycle 3, respectively.

**Table 3 foods-09-01314-t003:** CIELAB values of fresh calafate juice and cryoconcentrate samples during storage.

Day	Sample	L*	a*	b*	ΔE*
0	Fresh juice	33.95 ± 0.61 ^a^	31.21 ± 2.72 ^a^	3.11 ± 0.08 ^a^	-
	C1	21.71 ± 1.01 ^b^	39.58 ± 0.47 ^b^	2.68 ± 0.24 ^b^	14.84 ± 1.07 ^a^
	C2	10.41 ± 0.51 ^c^	43.40 ± 0.34 ^c^	2.32 ± 0.21 ^b,c^	26.52 ± 0.34 ^b^
	C3	7.22 ± 0.19 ^d^	46.71 ± 0.29 ^d^	1.71 ± 0.08 ^d^	30.92 ± 0.30 ^c^
7	Fresh juice	32.05 ± 0.09 ^a^	33.71 ± 1.00 ^a^	3.04 ± 0.05 ^a^	-
	C1	16.85 ± 0.23 ^b^	41.44 ± 0.29 ^b^	2.02 ± 0.02 ^b^	17.08 ± 0.31 ^a^
	C2	8.37 ± 0.14 ^c^	44.85 ± 0.25 ^c^	1.41 ± 0.13 ^c^	26.22 ± 0.24 ^b^
	C3	5.49 ± 0.26 ^d^	48.45 ± 0.28 ^d^	0.99 ± 0.03 ^d^	30.44 ± 0.26 ^c^
14	Fresh juice	31.12 ± 0.15 ^a^	34.07 ± 0.13 ^a^	3.00 ± 0.01 ^a^	-
	C1	16.05 ± 0.05 ^b^	41.94 ± 0.08 ^b^	1.65 ± 0.06 ^b^	17.05 ± 0.06 ^a^
	C2	7.97 ± 0.03 ^c^	45.07 ± 0.07 ^c^	1.32 ± 0.01 ^c^	25.68 ± 0.04 ^b^
	C3	4.96 ± 0.02 ^d^	49.18 ± 0.06 ^d^	0.91 ± 0.02 ^d^	30.28 ± 0.03 ^c^
21	Fresh juice	25.84 ± 0.82 ^a^	37.36 ± 0.32 ^a^	2.79 ± 0.05 ^a^	-
	C1	13.10 ± 0.60 ^b^	43.41 ± 0.35 ^b^	1.17 ± 0.04 ^b^	14.20 ± 0.45 ^a^
	C2	6.37 ± 0.16 ^c^	46.48 ± 0.35 ^c^	1.06 ± 0.06 ^c^	21.58 ± 0.27 ^b^
	C3	3.10 ± 0.05 ^d^	50.46 ± 0.31 ^d^	0.75 ± 0.05 ^d^	26.32 ± 0.17 ^c^
28	Fresh juice	20.25 ± 0.30 ^a^	39.60 ± 0.33 ^a^	2.33 ± 0.27 ^a^	-
	C1	9.49 ± 0.14 ^b^	45.30 ± 0.27 ^b^	0.92 ± 0.04 ^b^	12.26 ± 0.16 ^a^
	C2	5.04 ± 0.07 ^c^	48.31 ± 0.33 ^c^	0.85 ± 0.04 ^c^	17.58 ± 0.20 ^b^
	C3	1.97 ± 0.02 ^d^	52.34 ± 0.28 ^d^	0.50 ± 0.05 ^d^	22.36 ± 0.18 ^c^
35	Fresh juice	13.57 ± 0.45 ^a^	41.73 ± 0.24 ^a^	1.99 ± 0.04 ^a^	-
	C1	6.13 ± 0.02 ^b^	47.67 ± 0.46 ^b^	0.53 ± 0.03 ^b^	9.63 ± 0.28 ^a^
	C2	3.69 ± 0.15 ^c^	51.00 ± 0.41 ^c^	0.33 ± 0.03 ^c^	13.65 ± 0.38 ^b^
	C3	0.51 ± 0.37 ^d^	55.00 ± 0.26 ^d^	0.06 ± 0.01 ^d^	18.72 ± 0.37 ^c^

Different letters in the same column show significant differences at 5% between homogeneous groups in each variable to a LSD. C1, C2 and C3 represents cycle 1, cycle 2 and cycle 3, respectively. “-” corresponds to a control sample on the day of analysis.

**Table 4 foods-09-01314-t004:** Total bioactive compounds (TBC) values of fresh calafate juice and cryoconcentrates during 35 days of storage.

Day	TPC (mg GAE/g d.m.)				TAC (mg C3G/g d.m.)				TFC (mg CE/g d.m.)			
	Fresh Juice	C1	C2	C3	Fresh Juice	C1	C2	C3	Fresh Juice	C1	C2	C3
0	54.72 ± 0.02 ^a,A^	63.34 ± 0.66 ^b,A^	106.88 ± 0.77 ^c,A^	147.20 ± 0.04 ^d,A^	41.19 ± 0.07 ^a,A^	46.65 ± 0.65 ^b,A^	77.32 ± 0.48 ^c,A^	101.92 ± 0.53 ^d,A^	31.89 ± 0.45 ^a,A^	35.11 ± 0.58 ^b,A^	58.52 ± 0.34 ^c,A^	76.89 ± 0.41 ^d,A^
7	57.89 ± 0.20 ^a,B^	69.18 ± 0.38 ^b,B^	115.34 ± 1.25 ^c,B^	163.05 ± 2.30 ^d,B^	42.93 ± 0.50 ^a,B^	50.02 ± 0.40 ^b,B^	82.16 ± 0.59 ^c,B^	110.55 ± 5.35 ^d,B^	33.46 ± 1.29 ^a,B^	37.97 ± 1.05 ^b,B^	63.06 ± 0.55 ^c,B^	85.11 ± 0.08 ^d,B^
14	52.61 ± 0.38 ^a,C^	61.80 ± 0.31 ^b,C^	105.70 ± 4.34 ^c,C^	146.18 ± 3.73 ^d,C^	38.35 ± 0.59 ^a,C^	44.35 ± 0.21 ^b,C^	75.59 ± 0.53 ^c,C^	100.85 ± 4.28 ^d,C^	28.79 ± 0.52 ^a,C^	32.35 ± 1.27 ^b,C^	56.17 ± 0.56 ^c,C^	74.99 ± 1.25 ^d,C^
21	49.86 ± 0.39 ^a,D^	59.62 ± 0.39 ^b,D^	103.74 ± 0.97 ^c,D^	144.93 ± 0.77 ^d,D^	35.19 ± 0.66 ^a,D^	41.51 ± 0.73 ^b,D^	72.85 ± 0.06 ^c,D^	99.34 ± 3.59 ^d,D^	25.40 ± 0.51 ^a,D^	29.01 ± 0.55 ^b,D^	52.04 ± 0.34 ^c,D^	70.01 ± 0.64 ^d,D^
28	46.68 ± 0.17 ^a,E^	56.67 ± 1.09 ^b,E^	101.81 ± 1.79 ^c,E^	143.24 ± 1.23 ^d,E^	31.93 ± 0.44 ^a,E^	38.30 ± 0.18 ^b,E^	69.66 ± 0.12 ^c,E^	96.70 ± 0.93 ^d,E^	21.43 ± 0.39 ^a,E^	25.54 ± 0.27 ^b,E^	45.44 ± 0.85 ^c,E^	62.69 ± 1.08 ^d,E^
35	42.80 ± 0.56 ^a,F^	53.52 ± 1.85 ^b,F^	99.52 ± 1.64 ^c,F^	141.18 ± 1.71 ^d,F^	28.61 ± 0.31 ^a,F^	35.07 ± 0.47 ^b,F^	66.48 ± 0.38 ^c,F^	93.86 ± 4.29 ^d,F^	17.96 ± 0.28 ^a,F^	20.56 ± 1.23 ^b,F^	40.19 ± 0.06 ^c,F^	57.88 ± 0.80 ^d,F^

a–d: Different small letters in the superscript in the same row denote differences at 5% between the fresh calafate juice and their cycles, according to the LSD test. A–F: Different capital letters in the superscript in the same column denote differences at 5% in the sample during storage time, according to the LSD test. C1, C2 and C3 represents cycle 1, cycle 2 and cycle 3, respectively.

**Table 5 foods-09-01314-t005:** Total antioxidant activity (TAA) values of fresh calafate juice and cryoconcentrate samples during storage.

Day	Sample	DPPH *	ABTS *	FRAP **	ORAC **
0	Fresh juice	6.86 ± 0.51 ^a^	14.74 ± 2.73 ^a^	23.01 ± 1.87 ^a^	21.40 ± 1.25 ^a^
	C1	17.28 ± 1.47 ^b^	38.91 ± 3.23 ^b^	43.95 ± 3.04 ^b^	63.77 ± 3.79 ^b^
	C2	26.61 ± 1.51 ^c^	57.04 ± 1.11 ^c^	67.19 ± 2.57 ^c^	87.95 ± 1.22 ^c^
	C3	35.61 ± 2.04 ^d^	75.47 ± 7.34 ^d^	93.19 ± 5.74 ^d^	113.21 ± 5.96 ^d^
7	Fresh juice	7.11 ± 0.91 ^a^	15.25 ± 0.77 ^a^	23.91 ± 2.12 ^a^	22.10 ± 1.77 ^a^
	C1	18.36 ± 1.30 ^b^	41.42 ± 2.15 ^b^	46.64 ± 1.41 ^b^	68.06 ± 1.14 ^b^
	C2	28.95 ± 1.74 ^c^	61.71 ± 4.39 ^c^	72.51 ± 5.01 ^c^	95.20 ± 2.65 ^c^
	C3	39.51 ± 1.62 ^d^	82.92 ± 2.55 ^d^	102.23 ± 5.39 ^d^	124.12 ± 3.24 ^d^
14	Fresh juice	6.12 ± 0.25 ^a^	12.02 ± 0.37 ^a^	17.97 ± 1.71 ^a^	15.96 ± 2.11 ^a^
	C1	16.23 ± 0.91 ^b^	35.41 ± 1.84 ^b^	37.21 ± 3.27 ^b^	52.18 ± 4.16 ^b^
	C2	25.36 ± 1.01 ^c^	53.62 ± 2.41 ^c^	59.32 ± 4.91 ^c^	74.63 ± 2.54 ^c^
	C3	34.57 ± 2.21 ^d^	72.36 ± 1.99 ^d^	85.70 ± 3.22 ^d^	101.12 ± 5.37 ^d^
21	Fresh juice	5.33 ± 0.33 ^a^	10.98 ± 0.78 ^a^	16.34 ± 1.41 ^a^	14.42 ± 1.24 ^a^
	C1	15.08 ± 1.41 ^b^	33.08 ± 1.02 ^b^	35.28 ± 4.34 ^b^	49.26 ± 2.04 ^b^
	C2	24.24 ± 1.07 ^c^	50.99 ± 2.01 ^c^	56.76 ± 3.80 ^c^	71.88 ± 1.95 ^c^
	C3	33.41 ± 2.57 ^d^	69.15 ± 2.07 ^d^	83.43 ± 4.51 ^d^	97.95 ± 3.35 ^d^
28	Fresh juice	4.69 ± 0.40 ^a^	9.81 ± 0.81 ^a^	13.80 ± 1.71 ^a^	12.36 ± 1.23 ^a^
	C1	13.26 ± 1.02 ^b^	29.14 ± 2.17 ^b^	31.51 ± 2.36 ^b^	42.99 ± 3.04 ^b^
	C2	22.33 ± 2.14 ^c^	45.48 ± 1.06 ^c^	51.04 ± 3.25 ^c^	62.45 ± 2.06 ^c^
	C3	32.27 ± 0.43 ^d^	65.73 ± 1.94 ^d^	78.70 ± 2.74 ^d^	92.52 ± 4.33 ^d^
35	Fresh juice	4.11 ± 0.72 ^a^	8.54 ± 1.99 ^a^	12.04 ± 1.95 ^a^	10.69 ± 2.54 ^a^
	C1	12.33 ± 1.36 ^b^	26.25 ± 0.77 ^b^	28.49 ± 1.21 ^b^	37.85 ± 1.74 ^b^
	C2	20.96 ± 4.02 ^c^	42.11 ± 1.67 ^c^	45.38 ± 2.85 ^c^	57.12 ± 3.23 ^c^
	C3	30.22 ± 2.74 ^d^	60.96 ± 7.25 ^d^	71.18 ± 4.14 ^d^	82.16 ± 4.47 ^d^

Different letters in the same column show significant differences at 5% between homogeneous groups in each variable to a LSD. C1, C2 and C3 represents cycle 1, cycle 2 and cycle 3, respectively. * Free radical scavenging capacity and ** Ferric reducing antioxidant power. All were determined as (mM TE/g d.m.)

**Table 6 foods-09-01314-t006:** Pearson’s correlation coefficients (r) between biological active compounds content (TBC and TAA) obtained by centrifugal block cryoconcentration (CBCC) process (third cycle).

	TPC	TAC	TFC	DPPH	ABTS	FRAP	ORAC
**TPC**	1.00						
**TAC**	0.97 *	1.00					
**TFC**	0.96 *	0.96 *	1.00				
**DPPH**	0.94 *	0.99 *	0.98 *	1.00			
**ABTS**	0.90 *	0.98 *	0.99 *	1.00 *	1.00		
**FRAP**	0.91 *	0.97 *	0.98 *	0.99 *	1.00 *	1.00	
**ORAC**	0.90 *	0.96 *	0.97 *	0.98 *	0.99 *	1.00 *	1.00

* Significant at 5%.

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
