# Peer review of "Quality Attributes of Cryoconcentrated Calafate (Berberis microphylla) Juice during Refrigerated Storage"

_foods, 2020, doi:10.3390/foods9091314_

Round 1

Reviewer 1 Report

The manuscript entitled “ Quality attributes of cryoconcentrated calafate (Berberis microphylla) juice during refrigerated storage” need minor revision.

The results are interesting, but there are few things to improve

  • Description of antioxidant activity measured by different methods should be discussed more deeply (determine the correlation between these methods and between content of biological active compounds).
  • Statistical analysis – table 2 and 4 - please describe the meaning of capital letters and small letters – which are in columns, which in rows. Statistical significance should be more intuitive
  • a chapter 3 should be titled “results and discussion”, if the authors have combined them into one. There is no discussion chapter at this point… and conclusions should be 4
  • Please discuss in more detail the results of sensory evaluation - hence the "better taste, aroma" of the cryoconcentrated juice
  • how to explain the increase in TBC and TAA after 7 days of storage in all samples
  • how was the mean pH value determined for the samples – (see: http://goldbook.iupac.org/terms/view/G02621)

Author Response

RESPONSE TO REVIEWER COMMENTS

(Reviewer 1)

The manuscript entitled “Quality attributes of cryoconcentrated calafate (Berberis microphylla) juice during refrigerated storage” need minor revision.

The results are interesting, but there are few things to improve.

  • Description of antioxidant activity measured by different methods should be discussed more deeply (determine the correlation between these methods and between content of biological active compounds).

Thank you for this important point. A discussion about Pearson’s correlation coefficient test between the content of biological active compounds was added in the manuscript, with respective references (please see Section 3.6. Correlation between TBC and TAA, red letters). Additionally, information was added to the section 2. Materials and Methods (please see Section, 2.8. Statistical analysis, red letters).

  • Statistical analysis – table 2 and 4 - please describe the meaning of capital letters and small letters – which are in columns, which in rows. Statistical significance should be more intuitive.

     Thank you for the observation. The meaning of capital letters and small letters was rewritten as footnote in Table 2 and Table 4 (red letters).

  • A chapter 3 should be titled “results and discussion”, if the authors have combined them into one. There is no discussion chapter at this point… and conclusions should be 4.

Thank you for the observation. The titled in section 3 was rewritten as 3. Results and Discussion. The correct number in the section 4 was written as 4. Conclusions (red letters).

  • Please discuss in more detail the results of sensory evaluation - hence the "better taste, aroma" of the cryoconcentrated juice.

Thank you for the suggestion. A discussion was added in this section to clarify details of the results obtained from the sensory evaluation (please see Section 3.6. Sensorial analysis, red letters). In addition, two references were added to the manuscript, and all the numbers in the references section were modified (red numbers).

  • How to explain the increase in TBC and TAA after 7 days of storage in all samples.

Thank you for the question. The increase in TBC and TAA values during storage time have been attributed to the total soluble solids (TSS), since previous studies have indicated the release of glycans after some days under storage, causing the release of sugar, and consequently an increase in the TSS values, influencing the TBC and TAA content due to the direct relationship between the solids and the bioactive components. Thus, the subsequent decrease in TBC and TAA values ​​is due to the decrease in TSS values, since the TSS are consumed by microorganisms, reinforcing the relationship between TSS, TBC and TAA.

  • How was the mean pH value determined for the samples – (see: http://goldbook.iupac.org/terms/view/G02621)

Thank you for the observation. The pH value was determined individually, transforming each determination to the concentration units before the mean pH calculation. A line was added in the section to clarify some details in the pH determination (please see Section 2.3. Physicochemical analysis, red letters). In addition, one reference was added to the manuscript (red number), and all the numbers in the manuscript and references section were modified (red letters).

Dr. Patricio Orellana-Palma

Department of Biotechnology, Universidad Tecnológica Metropolitana, Las Palmeras 3360, P.O. Box, 7800003, Ñuñoa, Santiago, Chile.

E-mail address: p.orellanap@utem.cl

Reviewer 2 Report

The topic is interesting and the overall presentation of the study adequate. Since, however, the preservation of food is directly connected to the rate of each microbial decay, I would expect to see in the study some microbiological results.  

Specific comments

Lines 41-42 :  please provide some information regarding the seasonality of juice production

Lines 48-49 :  “due to…compounds” please rephrase since the meaning is not clear

What is the novelty of the study which makes it different from other previous studies? Please provide in the last paragraph of the introduction

Line 166: Were the jars sterilized?

Lines 250-251 : What are the expected products of the TSS consumption by microorganisms ? How does CBCC and storage affect the total microbial load of the juice? In case you do not have data please make a brief mention based on the literature since this is important in terms of consumption safety!

Line 279-281: Where do you attribute the decrease of pH during concentration?

Lines 289-296: Can the expected products of the TSS consumption by microorganisms lead to pH increase?

Lines 539-249: This is rather a summary than conslusions. Please rewrite pointing on the positive effects of CBCC application

Suggestions regarding language

Line 49 : “..and…rejection” → “ resulting to possible rejection by consumets”

Line 53: omit “Precisely”

Line 60:  please us comma instead of semicolon

Line 61 : “the easy” → “easier”

Line 72: “Therefore we aim..” → “The aim of the study was…”

Line 192: “produced” → “had”

Line 197: “be explained due…” → “be attributed to the centrifugation which…”

Line 274:” decay” “decrease”

Author Response

RESPONSE TO REVIEWER COMMENTS

(Reviewer 2)

The topic is interesting and the overall presentation of the study adequate. Since, however, the preservation of food is directly connected to the rate of each microbial decay, I would expect to see in the study some microbiological results.  

Specific comments

  • Lines 41-42:  please provide some information regarding the seasonality of juice production.

Thank you for the suggestion. The seasonality of juice production is directly related to the fruit production. A line about the seasonality of calafate production versus other berries was added (please see Section 1. Introduction, red letters).

  • Lines 48-49:  “due to…compounds” please rephrase since the meaning is not clear.

Thank you for the observation. The phrase was enhanced giving more details (please see Section 1. Introduction, red letters). Additionally, the phrase was coupled with comments from the other reviewer.

  • What is the novelty of the study which makes it different from other previous studies? Please provide in the last paragraph of the introduction.

Thank you for this important observation. A paragraph with the novelty of this study was added (please see Section 1. Introduction, red letters).

  • Line 166: Were the jars sterilized?

Thank you for the question. Only the jars (without juice or cryoconcentrates) were sterilized in order to avoid any type of contamination. The jars with juice or cryoconcentrate were not sterilized since the aim of the study was to investigate the stability "in natural form" of the cryoconcentrates (without heat treatment or the addition of stabilizers and/or preservatives) under refrigerated storage. Thus, to our knowledge, this is the first study to analyze the stability of cryoconcentrated samples over time.

A line with the sterilization step was added (please see Section 2.6. Storage study, red letters).

However, this question is interesting, since our research group has the idea of study the effects of heat treatments in concentrate juice and to compare with cryoconcentration, as well as to study the effect of sterilization on the cryoconcentrate juices. We just have to wait for the end of the COVID-19 pandemic quarantine.

  • Lines 250-251: What are the expected products of the TSS consumption by microorganisms? How does CBCC and storage affect the total microbial load of the juice? In case you do not have data please make a brief mention based on the literature since this is important in terms of consumption safety!

Thank you for the questions. For the first question, the literature indicates that the result of microorganism proliferation in fruit juices such as LAB and yeasts produce copious quantities of carbon dioxide and off-flavors. Thus, a paragraph was added in the section 3. Results and Discussion (please see Section 3.3. Physicochemical analysis, red letters). In addition, references were added to the paragraph, and all the numbers in the references section were modified (red numbers).

For the second question, a paragraph with information was added in the section 3. Results and Discussion with references (red numbers). (please see Section 3.3. Physicochemical analysis, red letters).

  • Line 279-281: Where do you attribute the decrease of pH during concentration?

Unfortunately, we do not understand the lines indicated by the reviewer, since the Lines 279-281 corresponds to a statistical description of the samples and their respective cycles and the effects of refrigerated storage on each sample (fresh juice and cryoconcentrates) (Lines 279-280). In the Line 281, the authors begin the description of the pH and TTA at day 0.

However, we understand that the reviewer possibly makes a mention to the CBCC effects on the pH and their decrease as the cycles advanced, which was mentioned in the next lines (please see Section 3.3. Physicochemical analysis, red letters).

  • Lines 289-296: Can the expected products of the TSS consumption by microorganisms lead to pH increase?

Thank you for this observation. The scientific literature indicates that TSS consumption by the microbial flora that proliferates in unpasteurized fruit juices can be attributed to the use of organic acids as energy source for microorganisms, a phenomenon that increases the pH together with the acid hydrolysis of various polysaccharides, in which the non-reducing sugars are transformed into reducing sugars (please see Section 3.3. Physicochemical analysis, red letters).

  • Lines 539-249: This is rather a summary than conclusions. Please rewrite pointing on the positive effects of CBCC application.

Unfortunately, we do not understand the lines indicated by the reviewer, since the Line 539 corresponds to Figure 3, and the Line 249 corresponds to TSS values. However, we understand that the reviewer possibly makes a mention to the conclusions, and that these should be rewritten on the positive effects of CBCC application.

Thus, the conclusions were rewritten with more concise details to results and advantages of CBCC application (please see Section 4. Conclusions, red letters).

Suggestions regarding language

Thank you for all the observations.

  • Line 49 : “..and…rejection” → “ resulting to possible rejection by consumers”

The line was changed (red letters).

  • Line 53: omit “Precisely”

The word was omitted.

  • Line 60:  please us comma instead of semicolon

The semicolon was changed by comma (red letters).

  • Line 61 : “the easy” → “easier”

The word was changed (red letters).

  • Line 72: “Therefore we aim..” → “The aim of the study was…”

The line was changed (red letters).

  • Line 192: “produced” → “had”

The word was changed (red letters).

  • Line 197: “be explained due…” → “be attributed to the centrifugation which…”

The line was changed (red letters).

  • Line 274:” decay” “decrease”

The word was changed (red letters).

Dr. Patricio Orellana-Palma

Department of Biotechnology, Universidad Tecnológica Metropolitana, Las Palmeras 3360, P.O. Box, 7800003, Ñuñoa, Santiago, Chile.

E-mail address: p.orellanap@utem.cl

Round 2

Reviewer 2 Report

The authors have responded to all comments and revised the MS adequately. A such the study is appropriate for publication in Foods.